# Airborne Microplastic in the Atmospheric Deposition and How to Identify and Quantify the Threat: Semi-Quantitative Approach Based on Kraków Case Study

**DOI:** 10.3390/ijerph191912252

**Published:** 2022-09-27

**Authors:** Kinga Jarosz, Rafał Janus, Mariusz Wądrzyk, Wanda Wilczyńska-Michalik, Piotr Natkański, Marek Michalik

**Affiliations:** 1Institute of Geological Sciences, Jagiellonian University, Gronostajowa 3a, 30-387 Kraków, Poland; 2Faculty of Energy and Fuels, AGH University of Science and Technology, A. Mickiewicza 30, 30-059 Krakow, Poland; 3Institute of Geography, Pedagogical University of Cracow, Podchorążych 2, 33-332 Krakow, Poland; 4Faculty of Chemistry, Jagiellonian University, ul. Gronostajowa 2, 30-387 Kraków, Poland

**Keywords:** airborne microplastics, urban pollution, microplastic pollution monitoring

## Abstract

Airborne microplastic is an emerging and widespread pollutant yet is still under-characterised and insufficiently understood. Detailed description of microplastic air pollution is crucial as it has been identified in human lungs and remote locations, highlighting the atmosphere as a medium of MP dispersion and transportation. The lack of standardization of methods for measuring and further monitoring of microplastic pollution is an obstacle towards assessment of health risks. Since the first recognition of MP presence in the atmosphere of Krakow in 2019, this research was conducted to further characterise and develop the methods for qualitative and quantitative analysis of airborne microplastic (attenuated total reflectance-Fourier transform infrared spectroscopy (ATR-FTIR); pyrolysis-gas chromatography–mass spectrometry (Py-GC–MS); scanning electron microscopy-energy dispersive spectroscopy SEM-EDS) and pre-treatment of samples. The data were gathered in seven cycles from June 2019 to February 2020. The methods used in the study allowed the identification and analysis of the changing ratio of the different types of synthetic polymers identified in the atmospheric fallout (low-density polyethylene, nylon-66, polyethylene, polyethylene terephthalate, polypropylene and polyurethane). Observations of interactions between microplastic particles and the environment were conducted with analyses of surface changes due to degradation. Different phases attached to the microplastics surfaces, with some of the inorganic contaminants transported on these surfaces determined also to be of anthropogenic origin. The methodology proposed in this study allows further characterisation of microplastic from multiple locations to provide highly comparable data, leading to identification of the sources of this phenomenon, as well as seasonal changes.

## 1. Introduction

Airborne microplastic (MP) is one of the most concerning yet one of the least described emerging pollutants in recent years. Lately, it has been detected in human lung tissue [1], making research on MP pathways, sources, concentration and more advanced characteristics even more urgent.

MPs are defined as particles composed of synthetic polymers of primary or secondary origin that are small enough to be easily dispersed in the environment [2,3,4]. Secondary MPs come from improper plastic waste management [5] and use of everyday products [6]. Plastics are exposed to destructive processes and agents (e.g., in washing machines) or environmental factors (e.g., UV radiation) [7].

It remains challenging to identify all the major sources and assess the proportions of the different sources of MP pollution [8,9]. MP is not a homogenous pollutant, and MP particles follow different pathways in the environment [10,11]. The number of studies on airborne MP is still limited [12]. Research on MP presence is widespread [13], with the main scope focusing on water and soil pollution [13,14,15]. MP presence has been confirmed in nature reserves [16], mountain ranges [17] and the Arctic [18].

Cities have features that increase the release rate of MP, such as anthropopressure, specific location, microclimate and urban morphology.

Interactions between MPs and the environment take place [19], resulting in heavy metal adsorption [20] and pesticide absorption into the surfaces of MPs [21]. Research is being conducted regarding toxicity of MPs so that their phytotoxic effects can be demonstrated [22], and increased antibiotic resistance [23], chemical composition and size of MPs play a key role regarding human health risks [24]. The current state of MP research does not reflect MP abundance in the environment [25]. There are studies regarding the negative impact of MP pollution on human health [26,27,28]. To assess health risks, complex and accurate data on airborne MP abundance are needed. The majority of studies on health impact are focused on extremes, such as workers in textile plants [29].

Data acquisition and development of standardised research procedures will enable monitoring of MP abundance in the atmosphere and determination of related risks with new standards for monitoring and understanding MP.

There are many methods in use in terms of sampling and examining MPs [30]; therefore, a unified procedure is crucial. Thus far, neither the sampling procedures nor the pre-processing and separation procedures have been standardised [31].

It is challenging to compare airborne MP abundance, which is connected to sample type: wet or dry deposition, and suspended atmospheric MP or MP settled in street dust. In some studies, no separation was carried out; in others, different solutions were used, most notably ZnCl_2_ for density-based separation [32] and H_2_O_2_ for digestion and later density separation with NaI [33,34]. In some studies, the researchers used filters either made of quartz [35], glass [36], polytetrafluoroethylene [13] or nitrocellulose [37]. Different methods generate different results in terms of the MP recovery rate. Differences originate from the different sampling methodologies (e.g., considering only particles of a certain size and using the optical method for hand-picking MP) and technicalities (e.g., different collectors, their location and height above ground level). The results acquired do not objectively regard MP abundance.

Classifying MP into categories (fibrous/non-fibrous, colour) [33,35,37] represents the main means of quantifying. Usually, from this count, only some of the particles are later tested with Raman spectroscopy, Fourier transform infrared (FTIR) or gas chromatography–mass spectrometry (GC–MS) techniques. This leads to the particle count being misleading [3]. Similarly, only part of the MP particles is identified, with the others assessed based on similarities. Data concerning the overall weight ratio of the different types of polymer ratios are of high value, complementing the results and helping to identify sources as MPs break down further in the environment.

The aim of this research is to recognize the major MP types and examine the changes in the chemical composition of such pollutants in the most unaltered form possible. Although the approach based on the counting of different particle types provides some useful information, the total relative mass of MP of a certain polymer type might not only be faster but could also be even more useful for a number of reasons. Further degradation and fragmentation of MPs take place. As a consequence, the gathered data can fail the objectivity criterion as MPs break down and separate over time, and one particle could easily become two or more. Decomposition, including nanoplastic emission, can be better understood and estimated by relative chemical composition data regarding MP abundance. Enhancing and validating MP separation protocols may be the best means of establishing the total net weight of different synthetic polymers in the most comparable and reliable manner.

There is ongoing research investigating the different degrees of MP toxicity. This pollutant often appears as a mixture of different synthetic polymers present simultaneously and should thus be examined as such [38]. More accurate research on health risks and toxicity might be carried out based on the relative chemical composition data.

The quantification of MP particles is even more problematic due to the limitation of the consideration of only particles of a certain size in the majority of cases in the literature [39]. Some measures have been taken to overcome this problem in comparing the results, yet a universal and reliable approach is lacking.

One of the promising techniques for fast and comparable airborne MP identification is coupled pyrolysis-gas chromatography–mass spectrometry (Py-GC–MS), which is now more frequently used to characterise MP in the environment [40], although primarily to characterise the particles individually, with the main methods used based on microscopic analysis combined with infrared or Raman spectroscopies. According to researchers, there is considerable room for improvement in the quality of quantitative analysis with Py-GC–MS [41], whereby the procedures require further development. The selection and separation of indicator ions has proven to be challenging when MPs exist in complex mixtures of biopolymers [41]. The results from different environmental matrices and places are important to fill the existing analytical gap [42].

The present study collected data from seven sampling cycles of atmospheric wet and dry deposition in an urban area of Krakow. The chemical composition of synthetic polymers was determined, and, additionally, the chemical analysis of the MP fibres’ surfaces was performed to provide new data to fill the gap in the area of potential health risks of MP inhalation.

## 2. Materials and Methods

### 2.1. Sample Collection and Processing

#### 2.1.1. Study Area

The city of Krakow lies in the south of Poland and is the second-largest city in the country, with a population of 779,115 for the period of sample collection (i.e., June 2019 to February 2020). In 2019 itself, it was visited by more than 14 million tourists. In terms of MP abundance, it is worth highlighting that the city is not only a place of culture and history but also of industry, with an increase in housing and other construction currently underway. The air quality in Krakow is one of the worst in Poland, and, moreover, among European countries (EEA Report No. 9/2020). Poor air quality is caused mainly by the high concentration of particulate matter [43] and nitrogen oxides [44]. The high concentration of pollutants is related, aside from local and distant emissions, to meteorological conditions and topography, and, in the case of Krakow, its location in the Wisla valley.

#### 2.1.2. Sampling

Passive sampling was chosen as the method for this study. Both dry and wet deposition were collected for the purpose of this research. A glass container (height: 0.21 m, collection area: 0.05675 m^2^) was placed on a platform on the rooftop of the five-storey-high building of the Pedagogical University in Krakow. It was situated on a platform to prevent contamination from the roof, itself at a total height of 35 m. The location for the container was chosen so that no vents would affect the collecting process. The sampling was carried out over eight months, from 2 June 2019 until 2 February 2020, in seven sampling cycles (cf. Table 1).

The first sample was collected over a two-month period to ensure sufficient amount of material for analysis. After analysis of the D68 sample, it was decided that one month would be sufficient to gather an optimal amount of material for the analytical techniques chosen.

#### 2.1.3. Sample Processing

For manual separation after collection, the sample container was closed and transported to a cleaned laboratory, where it was successively transferred to a Petri dish placed to evaporate on a laboratory water bath. The portion of material attached to the bottom and walls of the glass container was washed carefully using deionised water. The material was then the subject to visual inspection based on the physical properties according to the criteria described in Hidalgo-Ruz et al. [45] and using a binocular stereoscope. Fibres and particles that we had the slightest suspicion to be of synthetic polymer were picked with micro tweezers from the Petri dish with added deionised water. This allowed the smaller particles that would not have otherwise been picked to bind together. The initial picking of fibres and fragments was necessary as a majority (as per sample D68) or just some part of the sample contained materials of clearly biological or inorganic origin. The discarded particles were mostly atmospheric dust mineral grains, plant fragments and small insects.

Analysis of the sample prepared without this step was conducted, whereby the quality of the attenuated total reflectance-Fourier transform infrared spectroscopy (ATR-FTIR) spectra made further identification difficult and quantification impossible.

The sample from the final described sampling cycle (marked D0102) was concentrated using hydrofluoric acid on a cellulose filter instead of applying a visual inspection procedure. Sample D0102 was treated with 5% hydrofluoric acid. Then, it was shaken for a period of 1 h. Decantation commenced, followed by another hours of shaking in hydrofluoric acid. It was followed by rinsing the sample on a paper filter with distilled water until a neutral pH was achieved. It was then dried in a vacuum dryer at the temperature of 50 °C for a period of 24 h. Further, 2 mL of hydrofluoric acid was used for 100 mg of sample in a centrifuge tube.

#### 2.1.4. Quality Assurance

To ensure that the samples were not contaminated with external MPs, a cotton laboratory coat and non-synthetic clothes were worn, while the equipment and laboratory surfaces were wiped and rinsed, and plastic use was avoided in the protocol where possible. The laboratory staff were present in the laboratory only for the short periods necessary to perform the experiments (e.g., to transfer a new portion of the sample to a Petri dish). When possible, the containers and Petri dishes were covered.

To ensure quality, the following test was reconducted in the laboratory: the uncovered Anodisc (47 mm, 0.02 µm) filter was left for 12 h on the filtration set in the area of the experiment. The filter was then examined and singular potential microplastics were found.

### 2.2. MP Particles’ Identification, Quantification and Characteristics

#### 2.2.1. ATR-FTIR Spectroscopy

Mid-infrared spectra were collected in an ATR mode in the wave-number range of 400–4000 cm^−1^ using a Nicolet iS5 (Thermo Scientific Inc., Waltham, MA, USA) FTIR spectrometer equipped with an iD7 ATR accessory (Thermo Scientific) and DTGS detector. For each sample, 32 scans were acquired at a resolution of 4 cm^−1^. The ATR-FTIR analysis was performed for all samples. The obtained spectra were compared with those of the reference samples representing the six most popular synthetic polymers (i.e., polyethylene (PE), polypropylene (PP), polyurethane (PUR), nylon-66 (Nyl-66), polystyrene (PS) and poly(ethylene terephthalate) (PET)). Additionally, to verify the hypothesis of the presence of particles of rubber in the MP originating from grated tyres, the ATR-FTIR spectra of two types of commercial rubber were added (the typical tyre materials are marked T3 and T7).

#### 2.2.2. Py-GC–MS

Py-GC–MS was carried out using a pyrolyser unit (CDS Analytical, Oxford, PA, USA model 5200) coupled directly to a gas chromatograph (Agilent Technologies, Santa Clara, CA, USA, model 7890B) and a mass spectrometer (Agilent Technologies, model 5977A). The Py-GC–MS runs were carried out using small amounts (ca. 1.00 mg) of weighted sample placed in a quartz tube (L = 2.5 cm, ID = 1.5 mm) that was double-side plugged with quartz wool. Then, the sample was placed in a dedicated platinum coil of the pyrolysis unit, and the analysis procedure was commenced after the pyrolysis furnace was purged with helium (grade: 6.0) for 3 min. The pyrolysis tests were performed in a direct mode. Afterwards, the column was baked off at 260 °C for 5 min. The collected total ion chromatograms (TICs) were analysed using the deconvolution approach. The ion chromatograms were extracted from the TICs using Agilent G1034C MS, ChemStation software, Agilent Technologies, Santa Clara, CA, USA.

#### 2.2.3. Electron Microscope Observation of the MP Surfaces

A field emission scanning electron microscopy and energy depressive spectrometry (SEM-EDS) HITACHI S-4700 microscope equipped with a NORAN NSS energy dispersive spectrometer was applied to gather detailed information on the MPs in terms of the environment interaction. After visual inspection, typical synthetic polymer fibres from sample D68 were picked. Those fibres were of blue, black and orange colour. The selection criteria for those were colour, gloss and flexibility. MP particles were attached to the carbon holder and coated with carbon. A secondary electron signal was used for observation of the particles’ morphology. An accelerating voltage of 20 kV and beam current of 10°A were used for spot chemical analyses. This analysis was performed on chosen fibres from sample D68.

## 3. Results and Discussion

### 3.1. Atmospheric Deposition

The amount of material deposited (wet and dry) on the 0.05675 m^2^ surface was converted into a daily deposition per 1 m^2^, and the major differences were observed. The dynamics of the atmospheric deposition during the sample collection period were visibly different in summer in comparison with autumn and winter. June and July resulted in 0.006 g/m^2^/day, while August saw the highest deposition of 0.01 g/m^2^ day, and autumn and winter resulted in approximately half the amount of the daily deposition, with September, November, December and January resulting in 0.002, 0.003, 0.0025 and 0.003 g/m^2^/day, respectively. The biggest difference in weight of the daily deposited material occurred between August and September, whereby the August daily fallout weight was almost four times higher than that of September. The highest amount of deposited material was observed in the June and July period. The year 2019 was a record year for tourist traffic, and the June–August period is the most popular. Additionally, those months involve high vegetation growth and intense construction and maintenance work. Additionally, the sample mass is higher for months with higher wet deposition due to the presence of dissolved substances.

### 3.2. Visual Characteristics of MPs

Among the six samples prepared for further analysis, the transparent, blue, black, orange and red fibres were dominant in the anthropogenic material collected (Figure 1). There were, however, fragments that resembled torn pieces of foil, foam, irregular particles and some spheres that were potentially primary MP [46]. The material in all the samples looked similar. The number of particles considered to be MPs varied between the samples.

It should be emphasised that, based solely on the visual characteristics of the studied samples, the presence of MP could be confirmed. Some of the fibres were longer than 5 mm but still small enough to be transported into the atmosphere, and, therefore, they contributed to the overall airborne MP pollution.

The length-to-diameter ratio of the majority of the material present was greater than 3:1. This could pose a health risk as, the higher the ratio, the greater the likelihood of the MP entering the upper airways due to mucociliary clearance [29], while fibrous-shaped particles are generally considered more difficult to remove from the respiratory system [47].

Airborne MP presence has been confirmed so far in numerous cities in Asia [36,48], Europe [25,35] and North America [17]. Research on this subject does not follow unitary methods [49]. In some studies, the sampling methods are the passive collectors, suspended particulate samplers or even brushes. Not only sampling procedures but also pre-processing and separation procedures are various [50].

Quantitative analysis is sometimes based only on optical inspection [45]. In the present study, optical inspection was applied as the initial step and followed by instrumental analysis instead of quantification based on particles count.

In the collected samples (Figure 1), some of the particles present were clearly made of synthetic polymers, while others were possibly of natural origin (for instance, beetle carapace pieces) or from non-synthetic-polymer anthropogenic material, such as dyed natural silk, whose colourful fibres are often easier to recognise as MP, with most of the research suggesting that these are the major component of MP pollution [50].

Often, the polymer type is confirmed by instrumental methods for only a few representative MPs from the sample, and the classification is then extrapolated to the rest of the similar-looking particles. Some of the foreseen artificial-intelligence-based procedures of MP identification are based on the same principles [51]. However, data gathered in this way might be misleading or lacking in rigour. Although the samples collected had a similar appearance to the samples collected in 2018 [43], the analysis was more advanced in the current study.

### 3.3. Identification of MPs and Analysis of the Change in Their Relative Share over Time

#### 3.3.1. ATR-FTIR Identification

ATR-FTIR is a relatively fast technique suitable for the identification of synthetic polymers [48]. The ATR-FTIR spectrum of the D68 sample was selected for the clarification of our approach in terms of the qualitative analysis (Figure 2). All the acquired spectra were compared with the polymer and tyre rubber references. The presence of certain polymers in the MP samples was evidenced based on the presence of the characteristic absorption bands with regard to these references.

The ATR-FTIR spectra were analysed in detail with regard to the plastic references (Figure 2). The chosen synthetic polymers can be identified mainly by the specific spectra of functional groups [52,53,54]. In the spectrum recorded for Nyl-66, the bands at ~3300^−1^ and ~3200 cm^−1^, as well as at 1450 cm^−1^ and ~750 cm^−1^, can be recognised as the stretching, deformation and wagging vibrations of N–H bonds, respectively. The stretching, asymmetric deformation and wagging modes of the NH amide groups are present at ~1550 cm^−1^ and ~1650 cm^−1^.

The spectrum for low-density polyethylene (LDPE) features a characteristic doublet at 2800–3000 cm^−1^, ascribed to the stretching vibrations of methylene C–H bonds, bending at ~1470 cm^−1^ and ~1460 cm^−1^, and the rocking deformations of CH_2_ 710 cm^−1^.

In the case of PS, the presence of absorption at 3020 cm^−1^ may be assigned to the aromatic C–H stretch, 2850 cm^−1^ aliphatic C–H stretch and 1600 cm^−1^ of C=C bonds originating from the aromatic rings’ stretching vibration. The absorption at 1490 cm^−1^ ascribed to aromatic ring stretch, 1450 cm^−1^ –CH_2_– bend, 1027 cm^−1^ aromatic C–H bend and an intensive mode at ~700 cm^−1^ of aromatic C–H out-of-plane bend are also specific for PS.

PET can be recognised by the presence of C=O carbonyl group stretch (~1700 cm^−1^), C(O)–O stretching of ester groups (~1240 cm^−1^) and C–O stretch (~1090 cm^−1^) and aromatic C–H out-of-plane bend (~720 cm^−1^), together with the –CH_2_– deformation band (~1410 cm^−1^). For PUR, weak symmetric and asymmetric stretching vibrations of –CH_2_– aliphatic groups at 2970 cm^−1^ and 2870 cm^−1^ can be identified. Furthermore, there are stretching vibrations of the C=O carbonyl group at 1700 cm^−1^, the most intense peak at 1600 cm^−1^ from the stretching skeletal vibrations of C=C bonds present in the aromatic rings, stretching and bending vibrations of the N–H as a strong band at 1500 cm^−1^, deformation vibrations of the C–H at 1410 cm^−1^ and stretching vibration of C(O)O–C groups band at 1230 cm^−1^.

To identify the PP vibrations of CH bands at ~2920 cm^−1^, deformation vibrations of the plane methylene group in the spectral range of 1450–1480 cm^−1^, methyl groups’ vibrations in the 1370–1400 cm^−1^ range and the characteristic vibrations of the terminal unsaturated CH_2_ groups ~1170 cm^−1^, ~1000 cm^−1^ and ~850 cm^−1^ are recognised [55].

The ATR-FTIR spectra collected for all the studied MP samples are depicted in Figure 3. It may be conjectured that Nyl-66 and LDPE were present in all the samples. Amide bonds are also present in organic matter in the form of peptides (for example, in leaves, grasses and animals’ hair or fur) and, therefore, omnipresent in the environment. Other examined synthetic polymers, namely PS and PP, were recognised in samples D68 and D99. Sample D68 revealed only traces of PET. Based on the collected spectra, the presence of rubber particles in the tested MPs cannot be excluded (see the methyl and methylene bands at 2800–3000 cm^−1^). It should be noted that the spectra for samples D1011, D1112 and D1201, although similar to the rest of the MPs, show lower intensities, which is most likely due to a significant mineral content (likely silica and/or sulphates; see the bands at 1000–1300 cm^−1^. cf. Figure 2). For this reason, we decided to attempt to remove the mineral part by treating the MP samples with hydrofluoric acid (see Section Sample Pre-Treatment with HF). After mineralisation of sample D0102, the spectrum obtained was clear enough to identify PP, LDPE, Nyl-66 and PS presence.

The ATR-FTIR spectrum for sample D68 was the most varied among the studied MPs. This was surprising since the visual inspection indicated that colour or shape division provides limited or even misleading information about the MP particles’ origin. The passive MP collector was placed at the height of the fifth storey, which might be the reason behind the significantly lower presence of tyre traces even though they are considered as an important source of MP pollution [56]. Furthermore, the MP originating from tyres reaching this height might be small enough to be undetected. It can also be ascribed to another category because polyamide and PE fibres are also used for tyre manufacture [57], and, in this form, MP pollution originating from tyres could still potentially be present as, for example, nyl-66 fibre.

The presence of the most popular plastics of low- and medium-density MP [58] was observed. Identifying MP made with the most commonly produced synthetic polymers might be further expanded to other less prevalent polymers if the toxicity or health-risks data emerged. The ratio of different polymers’ total weight content might lead to improved understanding of the pathways and sources of this pollution. The presence of the main synthetic polymers produced might be a good base to develop and monitor the MP pollution index.

##### Sample Pre-Treatment with HF

An attempt was made to omit the manual concentration step after unsuccessful attempts to analyse the sample without any prior treatment due to the sediment (non-plastic matter) effect on the spectra. The previous six samples provided information about the MPs’ presence and characteristics, and demineralisation was performed with the final sample gathered through the use of HF solution (Table 1) and compared with the spectra from prior months. Therefore, a homogenic sample was obtained, for which a relatively good quality ATR-FTIR spectrum was acquired (Figure 3). As observed, this spectrum shows more features compared to the MP samples without HF treatment. However, one should note that the (likely) mineral-originating bands at 1000–1250 and 3000–3700 cm^−1^ are still observable in the spectrum. Therefore, although the HF pretreatment provides some improvement in the MP sample demineralization, a certain part of the inorganic components remain in the sample. Further study of this preparation path is needed.

#### 3.3.2. Py-GC–MS Study

##### A Semi-Quantitative Approach with Regard to the MP Component

A semi-quantitative analysis of the share of the six types of plastic most commonly found in the MP samples (i.e., PE, PP, PS, PET, nyl-66 and PUR) was feasible using the coupled technique of Py-GC–MS. Our approach is based on the MS analysis of the volatiles that evolved during the flash pyrolysis of the MP sample. Each type of plastic yields a certain characteristic set of volatile products when pyrolysed. It should, however, be kept in mind that the pyrolytic decomposition of MP is in fact the process of co-pyrolysis. It is well documented that, in such cases, the mechanisms (and products) of degradation may be mutually influenced. For this reason, we prepared a reference MP counterpart composed of the aforementioned six samples of plastic-taken equimassic (Figure 4). Then, the known mass of the reference mixture was subjected to the Py-GC–MS analysis at identical conditions to the real MP samples. Additionally, we performed the same analyses for all six plastics separately. Next, we analysed the collected total ion chromatograms (TIC) and extracted ion chromatograms (EIC). For each plastic type, we selected one characteristic and intensive mass line (*m*/*z*) that was unaffected by the mass lines contributed by other polymers’ decomposition. Integration of the chosen EIC peaks with regard to the mass of the real sample taken for analysis allowed us to calculate the semi-quantitative shares of each component of MP.

The Py-GC–MS confirmed the result of the ATR-FTIR analysis since the same types of polymers were found. Therefore, it was decided that it would be safe to rely solely on the Py-C–MS technique to recognise the synthetic polymers present in the atmospheric fallout. Moreover, use of this technique was successful in providing information about the relative concentration of the different polymer types (Table 2).

The results of Py-GC–MS confirm that the samples collected in June–July (D68), October (D1011) and November (D1112) were the most varied, with the MPs composed mainly of Nyl-66, LDPE, PS, PP and PET. The sample collected through September 2019 (D99) did not contain PET, while, in December, no PP was present, and the only synthetic polymers detected in August were Nyl-66 and LDPE.

In five samples, Nyl-66 accounted for the majority of MPs, comprising 56.9–93% of all the synthetic polymer mass. In the sample collected in October (D1011), LDPE was more prevalent than Nyl-66, but only slightly by 1.9%.

Nyl-66 is not only used for single-use-plastics but is also an important component in the automotive (up to 40% of total nylon usage in high-income countries), textile and construction industries [58]. Additionally, each sample contained a considerable amount of LDPE. This synthetic material is widely used in multiple industries due to its low cost of production, low weight and good flexibility [56]. LDPE is often used for packaging, carry-out and waste bags and agricultural and contracting films. The high content of LDPE may be likely ascribed on the one hand to its widespread use and on the other to its resistance to chemical degradation.

PS is commonly used for take-out packaging (e.g., coffee cup lids) and as expanded PS insulation foam [59]. It is also a very important material for construction applications in the form of expanded PS used for many purposes, primarily decorative tiles and mouldings, insulation blocks and as an additive to construction blocks [60]. In the five samples containing PS, its share difference reached 20%. Since PS is most often used and disposed of in expanded form, its extremely low density is understandable (from 0.01 g/cm^3^). This, in turn, allows for the uplift and transportation of larger secondary PS particles.

From sample seven (D0102), up to almost 50 wt.% of the polymer relative content is LDPE, although it is present in all the samples. LDPE is one of the most popular synthetic polymers used, and it is both durable and chemically resistant. The most common items made with LDPE are plastic bags, containers, toys, gas and water pipelines and high-frequency insulation.

Contrary to the fact that PET is a commonly used synthetic polymer, its amount in the samples was either very limited or none at all. One of the reasons behind this may be the fact that PET is more resistant to mechanical degradation in comparison to more durable synthetic polymer types [61]. Another reason is that PET has less chemical stability than other synthetic polymers. It is prone to some reactions, including acid and alkaline hydrolysis and methanolysis, which are also used in the chemical recycling of PET [62]. Those are possible reasons behind its limited presence in the samples. MP research still tends to focus on MP particles in soil, water and sediments. Airborne MP particles may, to some extent, form in different conditions, and, even if the degradation pathways are initiated in the same manner, the particles are present in a different medium. PET plastic is highly popular and used mostly as single-use plastic but also in textile products.

The lack of PUR particles in the studied MPs may be explained similarly. Observation of PUR behaviour under the influence of UV radiation (e.g., construction foams used for the sealing of cracks and holes in buildings) indicates that PUR ‘withers’ fast under direct sunlight.

Moreover, both PUR and PET are of higher density than other polymer types, at 1.20–25 and 1.38–1.41 g/cm^3^, respectively [63]. Their decreased presence or total absence at the height of a five-storey building might be explained by their density. It cannot be ruled out, however, that, apart from a clear difference between months, the main reason behind the presence of certain types of synthetic polymers and the lack of others may be caused by local levels of MP emission as opposed to the characteristics themselves.

The MPs detected are not only the most commercially used but are also the same as those present in the highest quantities in human lungs [1].

The main drawback of the Py-GC–MS method is the very limited amount of sample that can be subjected to analysis at any given time [64]. This, however, has little importance in the study of airborne MP since those particles, even in monthly periods of collection, do not amount to large volumes and, thus, do not pose a problem regarding sample size.

### 3.4. SEM Observations and Microanalysis of the Airborne MP Fibres’ Surfaces

Airborne MP fibres are subject to further degradation in the atmosphere (Figure 5). It is often assumed that fibrous MPs come from fibrous materials, such as textiles. With the use of SEM, images of the fibrous fragmentation of PE bottles collected on the pavement were documented. MP identification is often based on the protocol described by Chubarenko et al. [7] or others, where one of the criteria is that the fibres have to be equally thick throughout their entire length and not tapered at the end.

The phases identified on the surfaces vary considerably regarding chemical composition. The most prevalent elements are Al, Ca, Cl, Fe, K and Si. The presence of aluminium was noted in numerous locations. The presence of chlorine is connected to compounds with lower aluminium content. At the same time, a correlation between chlorine and sodium can be noted.

The chemical compositions of the substances attached to various MP fibres’ surfaces are depicted in Table 3, where SEM-EDS analysis led to the observation of aluminium salts’ attachment to airborne MP, which appears to be the most interesting observation made (Table 3). The main airborne aluminium source is human industrial activity [65], and, also, even though rare in a temperate climate, wind-transported soil particles. Aluminium salts detected in the samples are rarely present among air pollutants in Krakow [66,67]. Aluminium salts, among others, were previously detected as present in the form of particulate matter in the so-called ‘personal cloud’ [68]. The possible source of that is antiperspirant-deodorant products, where aluminium chlorohydrates are often used. There is a possibility for this to be direct proof that some MP fibres originated from clothes. Furthermore, airborne MP can not only represent a dangerous pollutant by itself but also a direct medium to contaminants from personal hygiene products. Other phases on the surfaces of MPs could be mostly classified as silicates or aluminosilicates. Metals such as iron and copper are present on the surface in complex compounds. Sulphur is also present as the only element or as a component in most of the analysed attached particles. The presence of this element might be associated generally with air pollution and wet deposition components. The exact positioning of the data subjected to EDS analysis can be found in the Appendix A.

## 4. Conclusions

Airborne MP pollution needs to be well recognised and addressed, especially in highly urbanised areas with significant human exposure. The proposed approach will lead to highly comparable results and, therefore, enable future monitoring and legislation. The presence of at least five different synthetic polymers in the atmosphere of Krakow was confirmed by means of both employed methods: ATR-FTIR spectroscopy and Py-GC–MS. The main synthetic polymers present in the atmospheric deposition in Krakow were Nyl-66 (47.5–93%) and LDPE (7–49.4%). PS was present in six samples (0.8–22.2%), PP in five (0.2–24.6%), PET in three (0.2–1.1%) and PUR was not detected. The absence of PUR and the lower concentration of PET can be connected to their higher density and reactiveness with the environment. Temporal changes in MP-type share were also confirmed, providing scope for further research on possible patterns and the reasons behind this. A wide spectrum of attached mineral phases, including aluminosilicates and aluminium salts, were observed on the MP fibres’ surfaces. In terms of ways to identify the MP particles’ source, further study of the presence on the surfaces of MP particles might lead at least to identification of the potential source thereof. For example, aluminium salts’ presence on surfaces might come from personal hygiene products. The inorganic pollutants on the surfaces of MPs potentially add to the risks associated with MPs’ inhalation. Py-GC–MS was efficient in airborne MP composition determination, and the results were consistent with the ATR-FTIR results. It might be possible to exclude the visual inspection step in future investigations, using instead the pre-concentration of MP via the demineralisation of samples using HF. However, this method still needs to be further validated.

## Figures and Tables

**Figure 1 ijerph-19-12252-f001:**
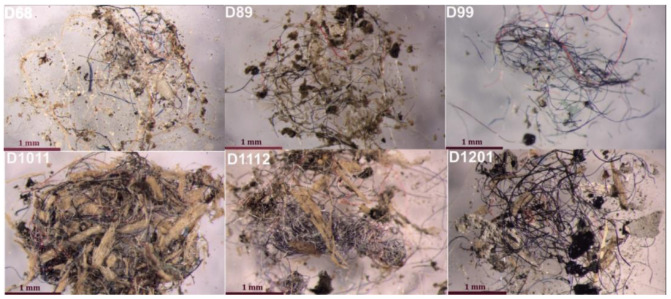
General view of the collected samples after visual identification and preconcentration.

**Figure 2 ijerph-19-12252-f002:**
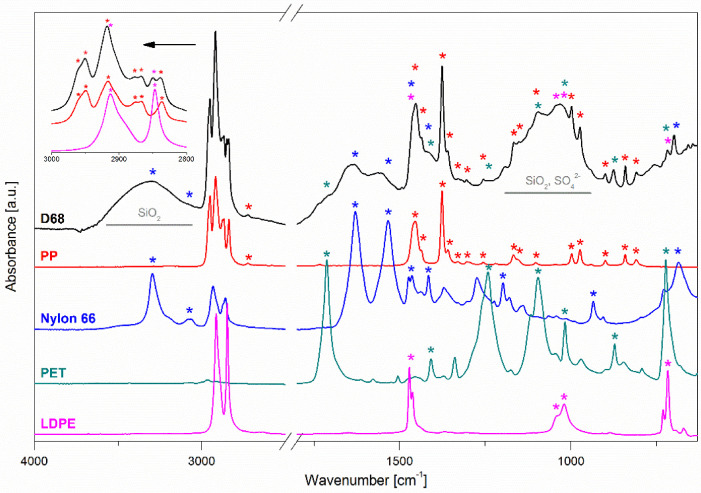
Methodology of the ATR-FTIR spectra interpretation: the case of the D68 sample. The colour asterisks above the D68 spectrum refer to the characteristic bands of certain plastic components of this sample.

**Figure 3 ijerph-19-12252-f003:**
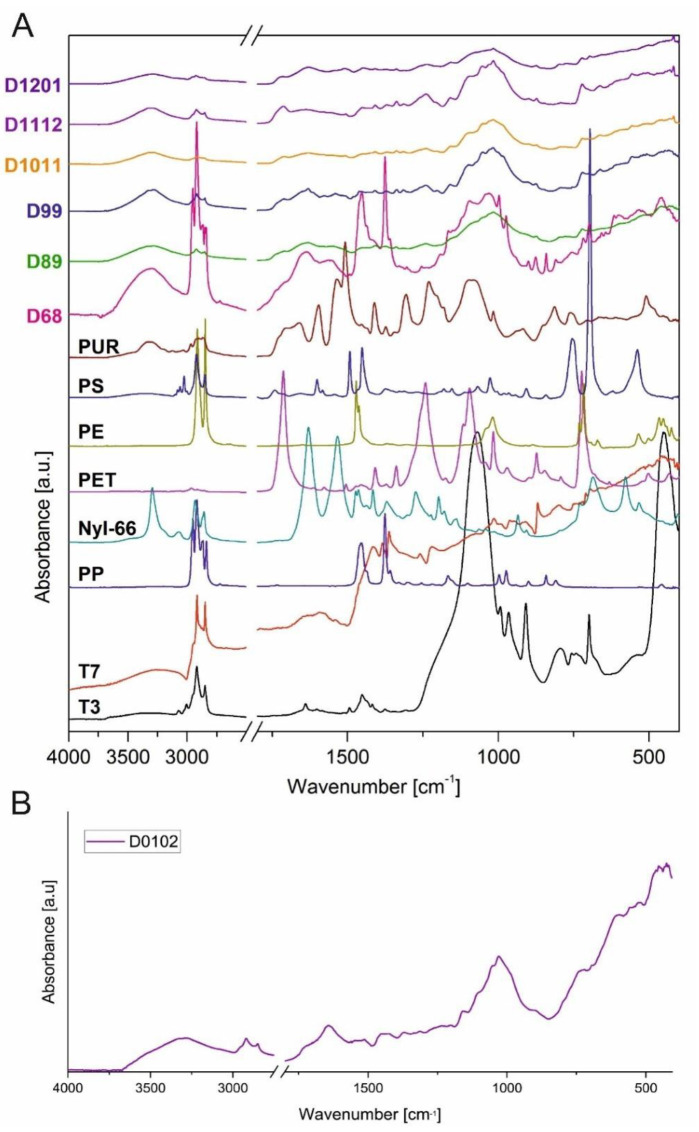
(**A**) ATR-FTIR spectra for the samples under study and all plastic and rubber references. (**B**) ATR-FTIR spectrum of the January sample; D0102 after HF (hydrofluoric acid) demineralisation.

**Figure 4 ijerph-19-12252-f004:**
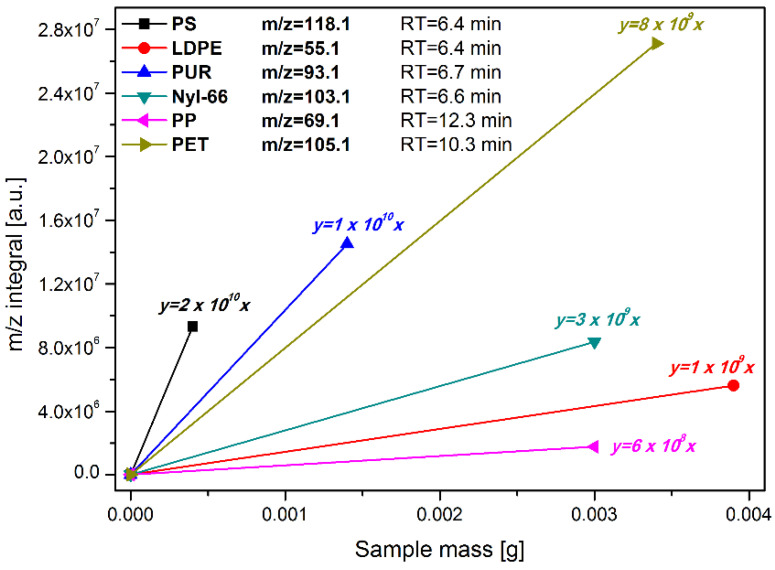
Calibration curves for the calculation of the contents of MP components.

**Figure 5 ijerph-19-12252-f005:**
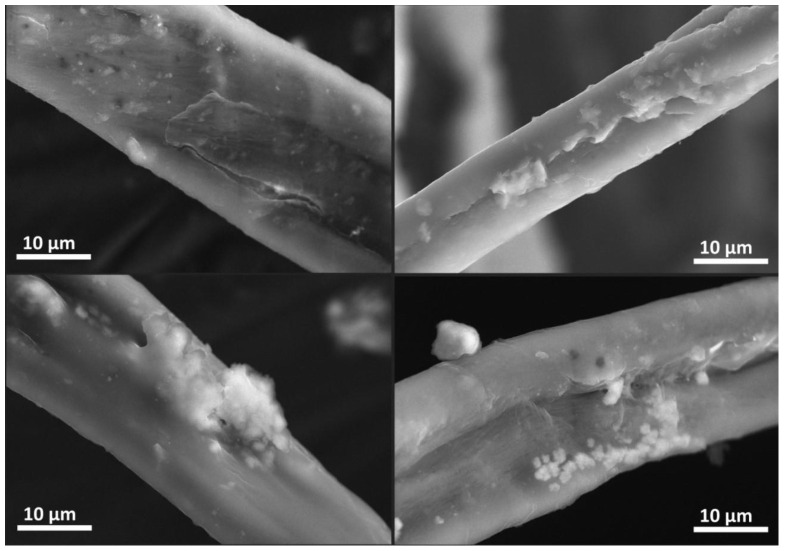
The SEM images of MP fibres with mineral phases attached (sample D68).

**Table 1 ijerph-19-12252-t001:** Summary table showing the sample data and implemented processing methods of samples, together with the techniques of their analysis.

SAMPLE	D68	D89	D99	D1011	D1112	D1201	D0102
Period of collection	2 June–9 August 2019	9 August–9 September 2019	9 September–1 October 2019	1 October–5 November 2019	5 November–3 December 2019	3 December 2019–3 January 2020	3 January–2 February 2020
Dry mass of total atmospheric deposition [g]	0.245	0.178	0.026	0.059	0.045	0.045	0.045
Daily average dry mass of atmospheric deposition [g/day]	0.0036	0.0057	0.0012	0.0017	0.0016	0.0016	0.0015
ATR-FTIR	X	X	X	X	X	X	X
Py-GC–MS	X	X	X	X	X	X	X
SEM-EDS	X						
Sample preparation procedure							
HF pre-treatment							X
Manual concentration	X	X	X	X	X	X	

**Table 2 ijerph-19-12252-t002:** Composition of the MPs by polymer type, determined by means of the analysis of the evolved pyrolytic gas (Py-GC–MS technique).

Polymer	Polymer Content [wt.%]						
	D68	D89	D99	D1011	D1112	D1201	D0102
PS	13.6	–	0.8	1.6	0.2	22.2	16.1
LDPE	14.7	7.0	17.7	49.4	9.6	13.8	21.9
PUR	–	–	–	–	–	–	
Nyl-66	66.9	93.0	56.9	47.5	89.7	64.0	34.8
PP	3.7	–	24.6	1.0	0.2	–	27.2
PET	1.1	–	–	0.4	0.2	–	

**Table 3 ijerph-19-12252-t003:** Chemical composition of substances attached to the MP fibres’ surfaces based on EDS analysis results [wt%].

*A*	*Orange MP Fibre Surface*							
*wt%*	OR 1	OR 2	OR 3	OR 4	OR 5	OR 6	OR 7	
**O**	35.35	16.74	n.d.	33.28	36.61	47.30	n.d.	
**Na**	2.90	10.18	n.d.	19.72	2.17	n.d.	5.84	
**Mg**	1.30	n.d.	n.d.	n.d.	1.65	2.19	n.d.	
**Al**	12.91	0.59	n.d.	0.73	8.49	0.66	76.50	
**Si**	28.59	1.81	n.d.	2.27	22.56	1.21	n.d.	
**S**	0.77	3.24	100.00	0.44	0.95	1.61	17.66	
**Cl**	4.89	39.34	n.d.	3.57	0.53	1.21	n.d.	
**K**	5.06	21.77	n.d.	24.63	2.25	0.75	n.d.	
**Ca**	1.27	6.33	n.d.	13.09	6.05	45.08	n.d.	
**Cu**	n.d	n.d.	n.d.	2.27	n.d.	n.d.	n.d.	
**Ti**	0.31	n.d.	n.d.	n.d.	n.d.	n.d.	n.d.	
**Fe**	6.25	n.d.	n.d.	n.d.	15.87	n.d.	n.d.	
**Zn**	0.42	n.d.	n.d.	n.d.	n.d.	n.d.	n.d.	
**Ba**	n.d	n.d.	n.d.	n.d.	0.93	n.d.	n.d.	
**P**	n.d	n.d.	n.d.	n.d.	1.91	n.d.	n.d.	
** *Total* **	100.00	100.00	100.00	100.00	100.00	100.00	100.00	
** *B* **	** *Blue MP Fibre Surface* **							
** *wt%* **	** *BLUE 1* **	** *BLUE 2* **	** *BLUE 3* **	** *BLUE 4* **	** *BLUE 5* **		** *BLUE 6* **	
**O**	46.19	47.60	43.81	14.58	18.10		42.51	
**Na**	1.66	7.14	9.74	1.52	n.d		4.01	
**Mg**	n.d	3.63	2.42	0.72	1.57		2.44	
**Al**	1.95	1.55	1.87	7.67	1.37		8.10	
**Si**	0.96	20.40	17.18	20.63	1.01		15.15	
**S**	18.24	2.67	2.86	3.57	1.72		3.87	
**Cl**	2.11	8.49	10.93	12.77	0.44		7.69	
**K**	1.25	5.24	6.81	11.59	0.42		5.02	
**Ca**	27.65	3.27	3.21	14.10	74.10		10.16	
**Cu**	n.d	n.d	n.d	n.d	n.d		n.d	
**Ti**	n.d	n.d	n.d	n.d	n.d		n.d	
**Fe**	n.d	n.d	1.17	12.87	n.d		n.d	
**Zn**	n.d	n.d	n.d	n.d	n.d		n.d	
**Ba**	n.d	n.d	n.d	n.d	n.d		n.d	
**P**	n.d	n.d	n.d	n.d	1.27		1.04	
** *Total* **	100.00	100.00	100.00	100.00	100.00		100.00	
** *C* **	** *Blue MP fibre surface* **							
** *wt %* **	**BK 1**	**BK 2**	**BK 3**	**BK 4**	**BK 5**	**BK 6**	**BK 7**	**BK 8**
** *O* **	35.35	28.25	31.19	27.74	32.31	43.27	22.11	20.29
** *Na* **	2.90	4.50	n.d	8.57	4.26	3.62	14.45	4.27
** *Mg* **	1.30	n.d	n.d	n.d	n.d	n.d	n.d	n.d
** *Al* **	12.91	16.07	38.77	16.03	1.28	0.49	1.20	13.79
** *Si* **	28.59	0.98	12.94	0.56	0.39	44.27	1.09	47.35
** *S* **	0.77	5.68	1.40	3.25	0.53	1.00	5.51	0.82
** *Cl* **	4.89	25.95	7.75	24.01	3.11	4.72	31.44	11.05
** *K* **	5.06	18.56	6.22	19.83	1.34	2.21	18.54	2.43
** *Ca* **	1.27	n.d	1.74	n.d	56.78	0.41	4.80	n.d
** *Cu* **	n.d	n.d	n.d	n.d	n.d	n.d	0.87	n.d
** *Ti* **	0.31	n.d	n.d	n.d	n.d	n.d	n.d	n.d
** *Fe* **	6.25	n.d	n.d	n.d	n.d	n.d	n.d	n.d
** *Zn* **	0.42	n.d	n.d	n.d	n.d	n.d	n.d	n.d
** *Ba* **	n.d	n.d	n.d	n.d	n.d	n.d	n.d	n.d
** *P* **	n.d	n.d	n.d	n.d	n.d	n.d	n.d	n.d
** *Total* **	100.00	100.00	100.00	100.00	100.00	100.00	100.00	100.00
** *wt%* **	**BK 9**	**BK 10**	**BK 11**	**BK 12**	**BK 13**	**BK 14**	**BK 15**	**BK 16**
** *O* **	33.79	n.d	47.76	9.48	20.94	14.08	n.d	n.d
** *Na* **	5.13	1.24	4.76	8.64	2.62	n.d	5.30	n.d
** *Mg* **	n.d	n.d	n.d	n.d	n.d	n.d	13.38	n.d
** *Al* **	12.02	16.39	16.83	33.97	37.07	63.14	27.33	68.93
** *Si* **	39.83	1.34	1.50	0.71	9.96	n.d	14.60	31.07
** *S* **	0.25	29.51	10.26	1.14	2.39	0.67	3.18	n.d
** *Cl* **	0.31	29.20	11.23	31.17	9.50	12.99	5.18	n.d
** *K* **	8.26	22.08	5.41	14.46	7.47	5.75	3.65	n.d
** *Ca* **	n.d	n.d	2.25	n.d.	7.82	1.10	27.39	n.d
** *Cu* **	n.d	n.d	n.d	n.d.	n.d.	2.27	n.d	n.d
** *Ti* **	0.14	n.d	n.d	0.44	n.d.	n.d.	n.d	n.d
** *Fe* **	n.d	n.d	n.d	n.d	1.50	n.d.	n.d	n.d
** *Zn* **	n.d	n.d	n.d	n.d	n.d	n.d.	n.d	n.d
** *Ba* **	0.28	n.d	n.d	n.d	n.d	n.d.	n.d	n.d
** *P* **	n.d	0.24	n.d	n.d	0.73	n.d.	n.d	n.d
** *Total* **	100.00	100.00	100.00	100.00	100.00	100.00	100.00	100.00

## Data Availability

Not applicable.

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
