# Peer review of "Airborne Microplastic in the Atmospheric Deposition and How to Identify and Quantify the Threat: Semi-Quantitative Approach Based on Kraków Case Study"

_ijerph, 2022, doi:10.3390/ijerph191912252_

Round 1
Reviewer 1 Report
see attachment

Author Response
Thank you for your time, please find the answers attached.

Reviewer 2 Report
The authors took a complete set of analysis methods to study microplastic pollution in a poor air quality area in Poland. This manuscript has valuable research subjects and the correct research methodology. Microplastic pollution in the air is still relatively poorly studied, and they are closely related to human health. Therefore, I believe that this manuscript is suitable for publication in IJERPH. However, there are many problems in the writing of this manuscript cannot be ignored, and the authors should conduct major revisions before reconsidering for publication.
1. I think it is inappropriate for the authors to use analysis methods as the main novelty highlight of this study. The methods used by the authors have all previously been used in microplastics research.
2. Use of abbreviations in the abstract is not appropriate.
3. The Introduction needs to be simplified. The authors were not needed to use 50 lines to describe the basics of microplastics. Try to use some review literature to briefly introduce the occurrence, toxicity, and effects of microplastics, such as:
10.1016/j.scitotenv.2021.150141
10.1016/j.envpol.2021.117999
10.1016/j.scitotenv.2021.150431
10.1016/j.chemosphere.2021.133146
4. For the reasons above, Line 124-128 needs to be rewritten.
5. Table 1, font. Besides, the information in the table is not clear enough, it is confusing.
6. Line 194, superscript.
7. Line 225, here puzzles me, did the authors do three tests of all the samples?
8. Line 257, this part of the discussion is lack of depth, not around the data, this is just some microplastic research has concluded by published studies.
9. Line 285, same problem, this part of the discussion simply shows only that the authors have identified the chemical composition of the microplastics. This is not an innovative analysis.
10. Line 520, show some specific values will be better.
Author Response
Thank you for your time, please find the responce in the attached file.

Round 2
Reviewer 1 Report
The revisions are not satisfying and not improve the paper quality
Author Response
Sir or Madam,
Due to no details (feedback) provided, according to Academic Editor instructions we decided to one more time revise the revision made, and provide additional changes and explanation according to your previous comments. Hopefully we corrected the mistakes and added data which will our manuscript fitted to your suggestions. Please find the additional changes and comments written in green i the PDF attached.

Reviewer 2 Report
My concerns are addressed.
Author Response
As in round 2 you stated that "your concerns are addressed", and we didn't obtain any new comments from you, there are no further answers to your revision. Thank you for your comments and suggestions made in the 1 round.